Original research

# Implementation of the national Getting it Right First Time orthopaedic programme in England: a qualitative case study analysis

Fiona Aspinal ![ORCID],[1] Jean Ledger ![ORCID],[1] Sarah Jasim ![ORCID],[2] Raj Mehta ![ORCID],[3] Rosalind Raine ![ORCID],[1] Naomi J Fulop ![ORCID],[1] Helen Barratt ![ORCID] [1]

¹Department of Applied Health Research, University College London, London, UK
²Care Policy and Evaluation Centre, The London School of Economics and Political Science, London, UK
³NIHR ARC North Thames Research Advisory Panel, Department of Applied Health Research, University College London, London, UK

**Correspondence to**
Dr Fiona Aspinal;
f.aspinal@ucl.ac.uk

## ABSTRACT

**Objective** To describe the implementation and impact of the Getting it Right First Time (GIRFT) national orthopaedic improvement programme at the level of individual National Health Service (NHS) Trusts.

**Design** Qualitative case studies conducted at six NHS Trusts, as part of a mixed-methods evaluation of GIRFT.

**Setting** NHS elective orthopaedic surgery in England.

**Participants** 59 NHS staff.

**Intervention** Improvement bundle, including bespoke routine performance data and improvement recommendations for each organisation, delivered via 'deep-dive' visits to NHS Trusts by a senior orthopaedic clinician.

**Results** Although all case study sites had made improvements to care, very few of these were reportedly a direct consequence of GIRFT. A range of factors, operating at three different levels, influenced their ability to implement GIRFT recommendations: at the level of the orthopaedic team (micro—eg, how individuals perceived the intervention); the wider Trust (meso—eg, competition for theatre/bed space) and the health economy more broadly (macro—eg, requirements to form local networks). Some sites used GIRFT evidence to support arguments for change which helped cement and formalise existing plans. However, where GIRFT measures were not a Trust priority because of more immediate demands—for example, financial and bed pressures—it was less likely to influence change.

**Conclusion** Dynamic relationships between the different contextual factors, within and between the three levels, can impact the effectiveness of a large-scale improvement intervention and may account for variations in implementation outcomes in different settings. When designing an intervention, those leading future improvement programmes should consider how it sits in relation to these three contextual levels and the interactions that may occur between them.

## BACKGROUND

Orthopaedic procedures such as total hip and total knee replacement are among the most common surgical procedures in the UK[1] and worldwide.[2 3] Demand is likely to increase given ageing populations and the associated

---

**STRENGTHS AND LIMITATIONS OF THIS STUDY**

⇒ This work is part of the first independent evaluation of the high profile, national improvement programme, Getting it Right First Time (GIRFT).

⇒ Observing 'deep-dive' visits to Trusts by the GIRFT team, reviewing local documentation, and interviewing clinicians and managers, enabled us to explore in-depth the factors that influenced the implementation and impact of GIRFT at a local level.

⇒ Although we only studied the process of implementation at six National Health Service provider Trusts, our case study approach created a rich and detailed picture of changes over time, in a variety of settings.

⇒ We spoke to a range of clinicians and managers at each site, but it is possible that we may not have captured all the improvements at each site, not least because the interviews took place several months after the first GIRFT visits creating a risk of recall bias.

---

increased need for arthroplasty surgery. This is likely to have a significant impact on health systems globally[4] including the National Health Service (NHS) in England, which already faces a backlog of planned surgery due to the COVID-19 pandemic.[5]

'Getting it Right First Time' (GIRFT) is one of the largest improvement programmes in the NHS. It began in orthopaedics in 2012.[6] Following government investment of over £62m, it now operates in 44 different specialties or clinical workstreams across England.[7] GIRFT aims to draw clinicians and managers together to reduce unwarranted variation, improve outcomes and maximise efficiency.[7] In orthopaedics, this includes changing practice (eg, prosthesis choice), organisational processes (eg, procurement, length of stay), and service delivery (eg, ring-fenced beds).[8] Clinical leadership is fundamental: GIRFT was established by a senior surgeon, who leads the orthopaedic workstream and chairs the wider

programme. Each workstream is chaired by a clinical lead from the relevant specialty. The programme includes components operating locally (ie, at NHS provider Trusts) and nationally (ie, across England). Local components include 'deep dive' visits to Trusts, while national components include reports describing how variations can be addressed. Before each visit, Trusts are sent a 'datapack', collated by GIRFT, describing their performance on over 100 variables, drawn from sources including Hospital Episode Statistics and the National Joint Registry. Datapacks cover use of evidence-based procedures and costs, and facilitate comparison to national and peer group averages. Discussion at the visits is driven by the datapack and tailored to the Trust. Attendees, comprising clinicians, managers and other relevant professionals, identify where improvements could be made. Although nationally there have been improvements in orthopaedic care over the last decade, these began before GIRFT. Other concurrent initiatives also targeted similar outcomes (eg, Lord Carter's review of NHS efficiency and NHS Right Care), so it is not possible to quantify the impact of GIRFT in isolation from other national programmes. Estimations of the additional impact of GIRFT deep dive Trusts visits, found a mix of positive and negative effects that were generally small compared with overall improvement trends.[9]

It is increasingly recognised that at an organisational level (eg, within individual NHS Trusts), the impact of an intervention such as GIRFT is shaped by multiple, contextual factors.[10] Context is defined as the underlying systems, culture and circumstances of the environment in which an intervention is implemented.[11] Contextual factors operate at three levels: microlevel (individual); mesolevel (organisational) and macrolevel (external/policy). At the microlevel of individuals in healthcare settings trying to implement change, collaboration must be fostered to facilitate change[12] and clinical leaders are often able to exert particular influence because of their professional credibility.[13] At the mesolevel, healthcare organisations themselves play a central role in improvement.[14] Relevant factors may include processes (eg, how strategy is developed); leadership (eg, how leaders engage with their external context) and underlying features (eg, leadership stability), all of which vary between different settings.[10] Outer 'macrolevel' context refers to influences beyond the organisation, such as government policy and regulation, which can have a strong influence on how organisations approach improvement.[15] While the intervention and local context interact to determine whether and how improvement occurs, it is increasingly recognised that interactions between the three levels of context can also serve to effect improvement or confound it.[10]

Although the influence of context on improvement is widely acknowledged, understanding of its role is still limited.[14] In this paper, we describe the implementation and impact of GIRFT at the level of individual Trusts. We explore both the contextual factors that influenced implementation and the ways in which the interaction of factors operating at different levels determined whether and how change took place. In doing so, we contribute to knowledge about how context influences how national improvement programmes effect change, and aim to inform the design and development of context-sensitive interventions in the future. This research was part of a wider mixed-methods evaluation of the GIRFT orthopaedic programme.[9 16]

## DESIGN

We used a case study approach to examine the different ways Trusts implemented GIRFT and how this influenced the impact of the programme locally. Case studies allow complex phenomena such as improvement to be studied in-depth where there are multiple variables and dynamics intersecting to influence change. The unit of the case study (here, organisations providing orthopaedic services) can be considered with respect to their internal context (such as professional leadership and organisational processes), and interactions between the case study and other external factors (eg, national trends or policy). Undertaking multiple case studies facilitates comparison and can increase the transferability of data to other situations and contexts.[17 18] Rather than data saturation, we sought breadth of perspectives. Decisions about the number of sites/interviews involved ongoing discussion within the research teams and a trade-off between research resources available and maximising the range of sites and the availability of interviewees (many of whom were front-line clinicians).

### Case selection

We identified six case study sites, all NHS Trusts participating in the GIRFT orthopaedic programme. We selected sites purposively to represent a spread of hospital types (district general and teaching hospitals); different levels of performance on GIRFT metrics and engagement with the GIRFT programme at the time of recruitment; and settings (region of England; rural vs urban).

This was an independent evaluation of the GIRFT programme. However, the GIRFT programme team facilitated our understanding of the programme by providing updates about programme changes, dates of deep-dive visits across Trusts in England, and access to key programme documentation. In July 2016, the GIRFT programme team provided a list of Trusts grouped according to their initial qualitative assessment of (1) performance on GIRFT metrics at that time and (2) extent to which they perceived managers and clinicians were committed to implementing change together. From this, we identified an initial list of six potential sites to meet the wider criteria and emailed the clinical lead for trauma and orthopaedics at each. Four responded positively, but we were unable to secure engagement at the other two—one high engagement and one low engagement site. We returned to the list, and identified and then recruited, two similar Trusts to replace to those in our initial shortlist.

The information provided by the GIRFT team to identify Trusts' level of engagement with the GIRFT programme at the early stage of implementation enabled us to draw on their knowledge of orthopaedic performance and key individuals at Trusts. However, the GIRFT team had no input into the final selection of sites (nor any other research decisions). The independence of this research from the GIRFT programme team was made clear to participating sites. Data collection was only carried out by members of the research team. The GIRFT programme team were not informed which sites or individuals took part, and all were anonymised throughout the study.

## METHODS

Case study research draws on a variety of data sources to construct a detailed picture of phenomena (in this case, implementation and impact of the GIRFT programme) in their 'natural setting'.[19] In each of the six sites, data were collected via non-participant observation of GIRFT visits, semi-structured telephone and in-person interviews, and collection and analysis of local literature/documents. Two researchers (JL and SJ) took the lead at different case sites (JL was the lead in four sites and SJ in two). Data were collected between October 2016 and May 2019. This long data collection period facilitated us conducting two rounds of interviews per case site and enabled the second of these, as far as possible, to coincide with Trusts' follow-up GIRFT deep-dive visits.

### Interviews

We (JL and SJ) conducted a round of semistructured interviews with orthopaedic surgeons, managers and other key staff who were present at the first Trust visit and/or knowledgeable about improvement work locally. The interviews covered: (1) organisational engagement with the GIRFT programme and the reasons for this; (2) practice and service delivery changes, particularly those concerning key GIRFT metrics such as ring-fenced beds, low volume surgeons, fixation methods and procurement; and (3) perceptions of the impact of GIRFT on any changes. The topic guide (see online supplemental file) was piloted prior to the interviews and refined iteratively as the study progressed to take account of emerging findings. A second round of interviews was conducted around the time of the Trusts' follow-up deep-dive visit. Similar topics (see online supplemental file) were covered, with a focus on change over time including implementation of GIRFT recommendations, and continued engagement with the programme. Some people were interviewed at both time points and some at a single time point. All interviewees gave informed consent. Interviews lasted between 20 min and 1 hour and were digitally recorded for professional transcription in full.

### Observation of deep-dive visits

Researchers (JL and SJ) attended and observed all GIRFT orthopaedic visits that took place at case study sites during the data collection period. This provided an understanding of the data being presented to the sites by the GIRFT team; the local context; and local receptiveness/engagement with the GIRFT programme. Researchers kept field journals for recording notes about attendee behaviours and interactions during meeting observations. They also attended deep-dive visits at non-case study Trusts and GIRFT events to better understand the wider programme and its delivery. This helped obscure the identities of participating sites from the GIRFT team. Consent to attend deep-dive visits was granted by the clinician leading the GIRFT programme in the case site, who asked attendees to indicate if they did not consent to researchers attending and taking notes. Consent was given at all case deep-dive visits. Researchers did not take fieldnotes when attendees indicated that a discussion point was confidential.

### Local documents

We collated documents from each site, including GIRFT data-packs and presentations, Trust annual reports and local documents about, for example, other improvement programmes or plans for local musculoskeletal or orthopaedic care pathways. These provided information about local approaches to improvement; competing/complementary priorities; the local population; Trust services and performance; and wider strategic decisions.

### Patient and public involvement

Patient and public advice on the design and conduct of this research was provided via the National Institute for Health and Care research Applied Research Collaboration North Thames' Research Advisory Panel. A patient representative (RM) was a core member of the research advisory group throughout the project.

## DATA ANALYSIS

Interviews were subjected to thematic analysis.[20] Data from documents and observations were reviewed to help interpret and contextualise interview data. Observation notes were included in thematic analysis where appropriate.

JL and SJ produced detailed, narrative case portraits for each case site, drawing on case transcripts, observation fieldnotes and documents. FA and HB read all data across all sites and, together with JL and SJ, held regular cross-case comparison analysis meetings to identify key themes across the data. Because each researcher has different case site and disciplinary expertise, we were able to identify and balance each individual's assumptions about the data. Together, we iteratively identified and determined the parameters of themes. FA then undertook line-by-line coding, applying these themes to data across all the sites. Findings were reviewed and confirmed by SJ, JL and HB.

The initial coding frame combined inductive and deductive approaches. It focused on whether and how sites engaged with GIRFT; reported 'improvements' in care; and the factors affecting this. Cross-case comparison

**Table 1** Participating case sites

| Criteria | Category | Counts |
|---|---|---|
| Type of Trust | District general | 3 |
| | Teaching | 3 |
| Setting* | Rural | 2 |
| | Urban | 3 |
| | Mixed | 1 |
| Level of engagement with the GIRFT programme at time of recruitment | High | 2 |
| | Medium | 2 |
| | Low | 2 |
| Region of England | East of England | 1 |
| | London | 1 |
| | North East & Yorkshire | 1 |
| | South East | 2 |
| | South West | 1 |

*Defined by interviewees and/or local public health annual reports.
GIRFT, Getting it Right First Time.

drew on the interaction between different levels of context (micro, meso and macro) to help explain how Trusts responded to a large-scale quality improvement programme.

Quotations used in the findings are illustrative of the points being made and representative of the wider patterns in the data.[21]

## FINDINGS

Information about participating sites, interviewees and data sources is presented before the findings from the case study analysis.

The six case sites represent a spread of Trust types, rurality, level of engagement with the programme and geographical region. To protect the anonymity of sites, we have summarised their characteristics data in table 1.

We interviewed 59 key stakeholders across the 6 Trusts, including orthopaedic surgeons (some of whom were also managers), other clinical staff and hospital managers (some of whom retained a clinical role), including at least one representatives of Trusts' executive leadership team in each site. Those interviewees with a joint clinical and managerial role, are referred to here as a 'hybrid manager'. Forty-six people were interviewed at time 1 and 13 at time 2. Some people preferred to be interviewed together with one or more colleagues: in site 1, one interview was conducted with two people and another with three, and in site 2 one of the interviews included two people. Table 2 shows the interviewees at each time point.

### Case study analysis

Before each 'deep dive' visit, Trusts were sent a 'datapack' from the national GIRFT team describing their performance on over 100 variables. Each visit was led by the GIRFT orthopaedic clinical lead. Discussion was driven by the datapack, so recommendations were specific to each Trust. Here, we explore first the factors that affected the adoption and implementation of GIRFT, including micro, meso and macro-level contextual influences. We then describe changes made to orthopaedic care at case study sites and the extent to which these were attributable to GIRFT.

### Microlevel contextual factors

At the individual level, managers and clinicians who were interviewed demonstrated good understanding of GIRFT and its aims. They described a variety of routes through which they first heard about the programme, including via

**Table 2** Interviewees per case site

| | Time | Surgeon | Nurse | Hybrid* | Manager | Commissioner | Total per time point | Total per site |
|---|---|---|---|---|---|---|---|---|
| Site1 | T1 | 1 | | 5 | 2 | 3 | 11 | **14** |
| | T2 | 1 | | 1 | | 1 | 3 | |
| Site2 | T1 | 2 | | 1 | 5 | | 8 | **11** |
| | T2 | 1 | | | 2 | | 3 | |
| Site3 | T1 | 3 | 1 | 1 | | | 5 | **7** |
| | T2 | | | 1 | 1 | | 2 | |
| Site4 | T1 | 2 | | 3 | 2 | | 7 | **9** |
| | T2 | 2 | | | | | 2 | |
| Site5 | T1 | 6 | | | 1 | | 7 | **8** |
| | T2 | | | | 1 | | 1 | |
| Site6 | T1 | 2 | | 2 | 4 | | 8 | **10** |
| | T2 | | | | 2 | | 2 | |
| Total | | 20 | 1 | 14 | 20 | 4 | T1=46 T2=13 | 59 |

*Hybrid includes clinician/managers.

the British Orthopaedic Association (BOA); colleagues; the initial GIRFT report[6] and attending deep-dive visits at their Trust. However, many claimed that neither the underlying concepts nor delivery approach were novel:

*Well, I suppose I became aware of the concepts that underpin GIRFT before GIRFT existed.* (Hybrid professional, Site 6)

Overall, interviewees across all sites supported what the programme sought to do—reducing unwarranted variation, maximising quality and promoting efficiency—and this made them more open to adopting the programme. As one participant said:

*Personally, I thought 'At last!' It is fantastic.* (Hybrid professional, Site 3)

Two major factors influenced how individuals perceived the intervention itself. The first was how they viewed the programme as a whole. At three sites, interviewees argued that their deep-dive visits focused on costs rather than quality. In one, for example, a senior manager argued that while the Trust always emphasised quality over cost, their deep-dive visit focused on:

*…money and efficiency and how many patients you can get through and how cheap you can buy prostheses for that work…* (Hybrid manager, Site 4).

In the sites where GIRFT was perceived to focus more on costs, there was low engagement, especially from surgeons, although in one site (site 6) this perceived cost focus triggered high engagement from managers.

The second factor was the approach taken by the GIRFT clinical lead. While almost all interviewees acknowledged his expertise, some described how his particular approach in deep-dive visits affected their opinion of, and willingness to engage with, the programme. Some admired his style for '*not tak[ing] any of [the surgeons'] rubbish*' (Manager, Site 2), others found him '*supportive*' (Surgeon, Site 5) and '*very compelling*' (Hybrid manager, Site 6). However, others considered him '*adversarial*' (Manager, Site 6) and '*patronising*' (Manager, Site 4).

Those interviewees aligned with the latter views expressed more reluctance to actively engage with GIRFT. Where these individuals had power to facilitate or resist engagement with the programme, these views negatively impacted adoption and implementation locally.

Equally, interviewees from all six sites argued that their visits had not taken the form of a two-way dialogue—a finding supported by the observations of the deep-dive visits (although changes to the deep-dive visit format in two sites later in the data collection period were observed to provide an opportunity for two-way information exchange)—limiting opportunities for attendees to question and challenge data, approaches and metrics. Consequently, at least some disengaged with the process, risking delivery of the programme. As one manager noted:

*I'm sure [the surgeons] would all value [the GIRFT] process and the information, but it has to be delivered in a way that's open to discussion and negotiation and mutual understanding.* (Manager, Site 4).

GIRFT is now being rolled out to over 40 other specialities, each of which has separate clinical lead. Interviewees who had attended visits in other specialities described different delivery approaches, which enabled two-way dialogue and a more flexible approach. This, they suggested, helped maintain their engagement.

The need for flexibility in implementing GIRFT recommendations was also highlighted. For example, interviewees felt that there was an expectation that Trusts would change practice simply to comply with GIRFT priorities, even where their outcomes were already above average and there was no cost benefit to derive from the change. For example, at sites 1, 2, 3 and 5, at least some surgeons were reluctant to switch to cemented fixation for all hip arthroplasty patients over 65 because it did not allow for surgeon responsiveness (site 5) or consider the financial or wider organisational consequences of the additional training that would be required for staff (site 1, 2, 3, 5).

The BOA's guidance[22] about implementing GIRFT recognised the need for some flexibility, and was welcomed by interviewees:

*Everybody in orthopaedics thinks that GIRFT is a good idea, but not everybody agrees with everything [the programme lead] says, so the [BOA] implementation document was a way of making [it] palatable to the wider profession.* (Hybrid manager, Site 3).

## Mesolevel contextual factors

At an organisational level, four of the sites had a long-embedded culture of quality improvement, including engaging with external quality initiatives in orthopaedics and across the Trust more widely. These Trusts already had internal processes for analysing performance data to inform service delivery and practice:

*…the staff are very pro-department and the Trust and they want to make improvements and make the best journey and the best outcomes for our patients. So they do come up with ideas of how we can improve. The consultants are also very proactive with regards to what improvements can be made and how the team can be best set up to get the best outcomes for the patients.* (Lead nurse, Site 1)

In several of the case sites, Trust managers commented that engagement in the GIRFT programme was useful because it provided a 'free, independent quality audit' (eg, Manager, Site 4) and 'benchmarking' (eg, Surgeon, Site 4). More importantly GIRFT analysed more local data than some had capacity or resources to analyse internally. Already being open to instigating evidence-informed change did not always lead to GIRFT-instigated change, but it could catalyse change processes so that results were experienced more quickly. For example, in case site 3,

GIRFT was used to expedite existing plans to streamline implant selection.

Nevertheless, across the sites, there was a feeling that GIRFT recommendations did not acknowledge the impact that Trust context could have on change. This included how the physical structure of a Trust could act as a barrier to improvement, even when individuals were convinced change was needed. For example, some interviewees argued that it was easier for multisite orthopaedic services to separate trauma and elective care and, in turn, to ring-fence beds. In contrast, single-site Trusts often struggled to set aside beds for elective orthopaedics, especially during winter months:

*So, like ring-fenced elective beds, that's an absolute critical requirement and recommendation of the GIRFT project. [Programme lead] talks about it a lot. And we all agree with that, but we still are unable to ring-fence elective orthopaedic beds because we are a massive hospital, an acute Trust, and lots of people are dying of flu and so on and so on. Those beds are needed to house the dying* [elderly] *in the winter.* (Surgeon, Site 5)

GIRFT recommendations linked to theatre capacity and bed usage, such as increasing the number of orthopaedic procedures per session or repatriating surgery from Any Qualified Providers, also encountered barriers. However, these were more about organisational complexity than organisational configuration. For example, interviewees talked about needing to '*vy[e] with urology, colorectal, vascular and all the other specialities'* (Hybrid manager, Site 6) to access theatre time, as well as negotiate bed availability across wards and arrange appropriate staffing.

Interviewees also noted that the complex planning needed to facilitate improvement in a busy acute Trust, means that change takes time. For example, interviewees in site 6 described it taking 8 months for ring-fenced beds to be implemented after Trust management had agreed because of other Trust priorities, space and bed demand-management, and associated cross-speciality negotiations (Surgeon, Site 6).

GIRFT's national peer benchmarking could, however, 'raise awareness' (Surgeon, Site 2) and 'leverage support' (Surgeon, Site 4) at a high level within Trusts that then catalyse improvement. As witnessed in site 6, GIRFT benchmarking data prompted the site to institute ring-fenced beds, something that had been requested by clinicians, but had been on the backburner at the Trust for some years.

### Macrolevel contextual factors

Implementation of GIRFT recommendations by Trusts was also impacted by factors outside the organisation. For example, establishing networks to manage complex and low-volume procedures could be hampered by complex dynamics at a regional level. This was especially the case where Clinical Commissioning Group boundary and hospital catchment areas and/or the regional sustainability and transformation partnership's (STP—now

reorganised as integrated care systems (ICS)) footprint were not aligned. In one site, GIRFT feedback was instrumental in a network being initiated, but this quickly broke down due to changing regional structures.

*[After the deep-dive visit] we straight away looked at forming networks amongst the hospitals [to organise] the low volume procedures. Unfortunately, it's broken down… So, that's why, politically, it broke down, because we had to align with our own STPs. So, politically speaking, we had to withdraw from that and try and start up a new alliance with* [other Trusts] *in region.* (Surgeon, Site 2)

### Impact

Interviewees were asked to describe any changes to orthopaedic care in recent years. Table 3 lists the changes that had been made and shows whether interviewees attributed this to GIRFT. Only two of the sites (sites 1 and 2) confirmed that the changes were a direct result of GIRFT. Indeed, many indicated that their Trusts were working towards or had already addressed some of the GIRFT priorities before the programme started in 2012. This had occurred in response to organisational or regional performance management processes (sites 1–6), organisational changes (eg, leadership changes, STP requirements or CQC status) (sites 1, 3 and 6) and/or preparing or delivering musculoskeletal pathway bids/contracts (sites 1, 2 and 6). Thus, while many changes aligned with GIRFT, few were actually attributable to the programme directly (see table 3).

However, where change was already planned or in progress, some participants found the GIRFT programme and deep-dive visits useful for formalising plans (eg, formalising the existing regional MDT network in site 5), catalysing action (network development in site 1), and leveraging support for change (eg, surgeons leveraging management support for ring-fenced beds in site 6).

In four of the sites, there was a question mark over the influence of GIRFT because interviewees were unsure of the relative contribution of the programme when other improvement initiatives were in play at the same time (sites 2, 3, 5 and 6). As one summarised about implant rationalisation:

*…because I have been pushing hard based on the GIRFT report, but a lot of colleagues across the UK and certainly here haven't been very keen to adopt the changes. But then with the tariff reduction, we have had no choice, so the combination of the GIRFT report and the tariff reduction, we have now made those changes.* (Hybrid manager, Site 3)

On average, sites reported that there were 10 other improvement initiatives (range 8–17) taking place at the same time as GIRFT, all impacting orthopaedic care. All sites reported at least one local monitoring and improvement initiative (eg, procedures rationalisation (site 2), infection minimisation (sites 1, 3–4), theatre utilisation (sites 1, 5–6), and procurement (sites 4, 6)); at least one relevant pathway or service redesign; and at least

**Table 3** Changes reported by participants in sites and their attribution to GIRFT

| Site | Recent change attributed to GIRFT by study participants | | | |
| --- | --- | --- | --- | --- |
| | **Fully** | **Partly** | **Possibly** | **Not** |
| 1 | Fixation method<br>Network development | | | |
| 2 | Fixation method<br>Network development | | | Procurement rationalisation<br>Arthroscopy reduction |
| 3 | | | Fixation method<br>Network development<br>Implant rationalisation | Length of stay |
| 4 | | | Procurement rationalisation | Fixation method<br>Low-volume surgeons<br>Network development<br>Infection rates |
| 5 | | | Ring-fenced beds | Procurement rationalisation<br>Low-volume surgeons<br>Network development |
| 6 | | Ring-fenced beds<br>Waiting list reduction | | Low-volume surgeons<br>Network development |

GIRFT, Getting it Right First Time.

two national efficiency and improvement programmes (eg, the Success Regime, Right Care, Cost Improvement Programme and Vanguard status). These could, as in the example above, work together to provider leverage for improvement.

## DISCUSSION

We have described the contextual factors that influenced adoption and implementation of GIRFT in orthopaedics and the ways in which these interacted to determine whether and how change took place. Contextual factors operated at three different levels, to varying degrees: the level of the orthopaedic team (eg, how individuals interacted with the intervention); the wider Trust (eg, competition for bed space and theatre time); and the health economy more broadly (eg, requirements to form local networks). Although all sites had made improvements to care in recent years, very few were a direct consequence of their GIRFT visits. GIRFT did, however, contribute to some improvements, although indirectly. Some used GIRFT to provide evidence, which helped to catalyse and formalise existing plans or processes, and to leverage cross-professional and organisational support. Along with other concurrent initiatives, GIRFT also helped create an environment receptive to change. It was particularly impactful where its recommendations and pre-existing Trust and professional priorities aligned. Unsurprisingly, where GIRFT measures were not a local priority because of more immediate demands—for example, financial and bed pressures—GIRFT was less likely to influence change. Programme leadership influenced acceptability and engagement with the programme both positively and negatively, and suggest that finding the right expertise and power balance between those leading and those

implementing change is essential when promoting large-scale change.

This is part of the first independent evaluation of GIRFT.[9 16] Our mixed-method approach enabled us to provide a comprehensive and robust understanding of the programme and explore its impact from different perspectives. Previous similar evaluations tend to focus on quantitative analyses. We only focused on GIRFT in orthopaedics but the findings about interactions between the intervention and multilevel contextual factors are transferable to other GIRFT clinical streams and other large-scale improvement programmes.

Although we have studied the process of implementation in a small number of sites (as is usual in this type of research), our case study approach led to a rich and detailed picture of changes as they occurred over time, in a variety of settings. The researchers collecting data (JL and SJ) were aware of the GIRFT programme team's initial engagement rating that we used to select Trusts. However, once case sites had been recruited, the focus of the evaluation was on participating Trusts'/individuals' perspectives on engagement with the GIRFT programme. As such, the initial assessment of engagement by the GIRFT team did not influence data collection or analysis. Although we identified common factors affecting engagement with GIRFT across the case sites, the small number of cases for each level of engagement means that we might not have identified the full range of factors that affect different, especially lower, levels of engagement with large-scale improvement programmes such as GIRFT. Two sites (one highly engaged and one with low engagement according to the GIRFT team's list) that were approached about taking part in the study did not respond to us and, consequently, we do not know their

reasons for choosing not to participate in this evaluation. At each site, we invited staff to interview who were present at the first GIRFT visit or otherwise knowledgeable about improvement work locally. We spoke to a range of professionals, including clinicians and managers, across all sites. However, it is possible that we have not captured all the changes that took place at Trusts, or all the factors that influenced implementation. It is also important to note that the recommendations to each Trust were bespoke, based on their performance data at the time, introducing a degree of heterogeneity into the content of the intervention. We collected empirical data between 2016 and 2019 while the programme was continuing to evolve, but we did not find any major changes during the evaluation period. Finally, the interviews took place several months after the first visits, creating a risk of recall bias.

Although there have been significant improvements in orthopaedic care nationally, many began prior to GIRFT. Changes observed over the past 10 years are likely attributable to both GIRFT and other concurrent initiatives, but it is not possible to determine their relative contributions. The additional impact of Trust visits was mixed.[9] The findings we describe here help explain why this is the case: where GIRFT aligned with pre-existing Trust and professional priorities, staff were able to leverage the programme to support change. Where GIRFT measures were not a local priority, it had much less impact. In 2020, the GIRFT team published an internal evaluation[23] of the orthopaedic workstream, citing individual case studies that exemplified success. Our diverse, purposively selected range of sites facilitated cross-case comparison. This paper also adds to the wider literature about the role of context in healthcare improvement. To date, only a small number of studies have explicitly focused on interactions between contextual factors.[24] We have demonstrated that dynamic relationships within and between levels, rather than any one factor individually, were key in shaping the impact of a major national improvement programme.

Interviewees from all our sites argued that their deep-dive visit had not been a two-way dialogue, with limited opportunities for attendees to question and challenge. Many also reported that GIRFT did not acknowledge the importance of local organisational context. Both things negatively influenced the impact of GIRFT. Those designing and delivering future improvement programmes should consider how all three levels of context (macro, meso and micro) and the interactions that may occur between them could affect their intervention, rather than focusing on individual factors or levels in isolation. This is especially important for large scale (eg, national) programmes where there is likely to be significant variation in local context. Case study research, with its small sample sizes, does not set out to create explanatory models for implementation. However, we highlight a number of contextual influences that are likely to be important, such as the nature and configuration of participating Trusts, and the presence or absence of external pressures, for example financial or regulatory challenges. Although some contextual factors can be modified by Trusts (eg, organisational cultures and leadership) many can not (eg, government policy) and, therefore, need to be taken into account by those implementing improvement programmes[24]

GIRFT is one of the largest improvement initiatives in the NHS. Following its introduction in orthopaedics, it was rolled out at substantial cost in over 40 other clinical areas, without independent evaluation. We have previously noted that changes to orthopaedic practice over the past 10 years are likely attributable to both GIRFT and other concurrent initiatives.[9] Our findings here help explain why the additional impact of GIRFT visits to individual Trusts was mixed: although all case sites had made improvements to care, very few were directly attributed to GIRFT. A range of factors, operating at different levels, also influenced sites' ability to implement GIRFT recommendations. As well as early engagement with rigorous evaluation design, those delivering future improvement programmes should consider how their intervention sits in relation to the three levels of context (macro, meso and micro) and the dynamic interactions that may occur between them. This is especially so given the advent of ICSs which, until they are embedded, might create more complexity, but which have the potential to improve information flow and collaboration as these multilevel contexts become more integrated and dynamic relationships between the contextual levels change.

**Acknowledgements** We thank Patricia Hallam for administrative support for the study. We thank all the other members of the research team who contributed to other parts of this mixed-methods evaluation. We thank the public members of the NIHR CLAHRC/ARC North Thames Research Advisory Panel who commented on all aspects of the study and the professional stakeholders who gave their time to take part in interviews and provide relevant documentation. We thank the GIRFT programme team for providing lists of sites from which to recruit case sites anonymously.

**Contributors** HB was principal investigator. HB and RR initiated the research. HB, NJF and RR designed the evaluation. SJ and JL were responsible for qualitative data collection; FA, JL, SJ, RM, NJF and HB analysed the qualitative data. FA, JL, SJ, RR, NJF and HB drafted the manuscript. All authors contributed to reviewing and substantive revision. All authors approved the final version. All authors had full access to all the data in the study and accept responsibility to submit for publication. HB will act as guarantor.

**Funding** This reports independent research funded by the National Institute for Health Research (NIHR) Collaboration for Leadership in Applied Health Research and Care (CLAHRC), North Thames. The NIHR Applied Research Collaboration (ARC) North Thames provided a costed extension (award/grant number is not applicable).

**Disclaimer** The views expressed in this publication are those of the authors and not necessarily those of the National Institute for Health and Care Research or the Department of Health and Social Care.

**Competing interests** None declared.

**Patient and public involvement** Patients and/or the public were involved in the design, or conduct, or reporting, or dissemination plans of this research. Refer to the Methods section for further details.

**Patient consent for publication** Not applicable.

**Ethics approval** This study involves human participants and was approved by Ethics committee: NHS North West - Liverpool East Research Ethics

CommitteeReference number: 16/NW/0654. Participants gave informed consent to participate in the study before taking part.

**Provenance and peer review** Not commissioned; externally peer reviewed.

**Data availability statement** No data are available. Due to the professional sensitivity of these data, qualitative data are not publicly available.

**ORCID iDs**
Fiona Aspinal http://orcid.org/0000-0003-3170-7570
Jean Ledger http://orcid.org/0000-0003-2523-7971
Sarah Jasim http://orcid.org/0000-0003-3940-6350
Raj Mehta http://orcid.org/0000-0002-4003-530X
Rosalind Raine http://orcid.org/0000-0003-0904-749X
Naomi J Fulop http://orcid.org/0000-0001-5306-6140
Helen Barratt http://orcid.org/0000-0002-1387-137X

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
