## [Reviewer comments · BMJ Open]

ARTICLE DETAILS

TITLE (PROVISIONAL)	Implementation of the national Getting it Right First Time orthopaedic programme in England: a qualitative case study analysis.
AUTHORS	Aspinal, Fiona; Ledger, Jean; Jasim, Sarah; Mehta, Raj; Raine, Rosalind; Fulop, Naomi; Barratt, Helen

VERSION 1 – REVIEW

REVIEWER	McLaughlin, Joanna University of Bristol, Translational Health Sciences
REVIEW RETURNED	12-Jul-2022

GENERAL COMMENTS	Manuscript ID bmjopen-2022-066303 entitled "Implementation of the national Getting it Right First Time orthopaedic programme in England: a qualitative case study analysis." for BMJ Open. Reviewer's comments Thank you for the opportunity to review this manuscript. Overall, I find the research to be of a good standard and the research question and setting highly relevant to the current context of demand on elective orthopaedic services. The main recommendation that intervention development in the future should place emphasis on understanding and accounting for micro, meso and macro contextual factors faced by the target organisations is important. Further consideration of the relevance of the changing organisational landscape with the advent of ICSs would strengthen the value of the recommendations. I would recommend that the authors provide clarification or additional information on the following points: What was the nature of the relationship between the research team and the GIRFT programme team? Collaboration is mentioned but it is not clear whether the evaluation was invited/commissioned and how much influence the programme team had. I note that elements of the evaluation data collection were facilitated by the programme team e.g., electronic surveys, and description of site engagement levels for purposive sampling. How was the figure of six decided upon for the sample size? The limitations of collecting data from only two sites deemed to have had low engagement with the GIRFT implementation are not
--

	discussed but are important. Was data saturation an element in the decision over the number of interviews conducted? The purposive sampling seems to have been successful in reaching a range of hospital types and settings. What was the nature of the Trusts that were approached to participate but where engagement was not achieved? This should be discussed in the limitations, particularly if there is the possibility that negative perception or experience with GIRFT was a significant factor in the Trust's lack of interest in engagement in the research. Little detail is provided on the analysis methodology. Were transcripts double coded and how were the individual researchers' analyses combined? The observation of the GIRFT visits seems a valuable part of the data collection but it is not clear what contribution to the analysis and findings this element provided. For example – the findings from the interviews include the statement that the GIRFT visits were not opportunities for two-way dialogue. Could the field journal notes be used to comment on this issue from the perspective of an observer? Comment is made that the interviews took place several months after the GIRFT visits – could more detail be provided on the timings of the interviews? The data collection window is wide (Oct 2016 to May 2019). The quotes on page 10, lines 55 and page 11 line 7 are distracting due to the use of the square brackets to correct '[b]ecause', and 'vy[e]'. Are these necessary? were they from written documents or typing mistakes in transcription from audio recording? Vie, rather than vye? The related paper: Barratt H, et al. BMJ Open 2022;12:e058316. doi:10.1136/bmjopen-2021-058316 suggests that "later cohorts had made changes prior to their visit because of information they gleaned from Trusts visited earlier in the process" was this explored in this qualitative work? Could the authors provide a view on researcher reflexivity in this study? (as per the SRQR supplementary document). This may need to take a particular focus on the impact of having observed the visits themselves ahead of the interview process. Did researcher knowledge of the GIRFT programme's categorisation of some of the sites as having had 'low engagement' shape the data collection and analysis? The findings present useful detail on the nature of the single clinical lead in this programme, the differences in perception of his style and contribution, and the potential impact of this role. These findings do not seem to have been explicitly translated into the recommendations; can the researchers provide any further recommendations specific to this insight? The consent process for the interviews is clear – detail is required on the consent situation for the visit observations.
--	--

VERSION 1 – AUTHOR RESPONSE

Response to reviewer’s comments

We would like to thank the reviewer for their comments and suggested clarifications. We have revised the manuscript in response to these comments and provide a summary of changes here. Where we have not addressed reviewer comments in the manuscript, we provide additional information here.

Reviewer comments	Response and action
(a) Further consideration of the relevance of the changing organisational landscape with the advent of ICSs would strengthen the value of the recommendations.	While these data were collected before the advent of ICSs, we have now added a sentence to help contextualise the findings/recommendations within the changing organisational landscape. We have added a comment on page 16 to address this point.
1. What was the nature of the relationship between the research team and the GIRFT programme team? Collaboration is mentioned but it is not clear whether the evaluation was invited/commissioned and how much influence the programme team had. I note that elements of the evaluation data collection were facilitated by the programme team e.g., electronic surveys, and description of site engagement levels for purposive sampling.	Further detail has been provided about the nature of the GIRFT programme’s team relationship with the research team. As the paper now indicates on page 5-6, this was an independent evaluation of the GIRFT orthopaedic programme, and all decisions were made by the researchers in the team. The GIRFT team did provide useful information to about the programme, dates of and reports from deep-dive visits to help facilitate the research team’s understanding of the programme.
2. How was the figure of six decided upon for the sample size? The limitations of collecting data from only two sites deemed to have had low engagement with the GIRFT implementation are not discussed but are important. Was data saturation an element in the decision over the number of interviews conducted?	Clarification has been provided in the text to indicate how the number of cases (and indeed, number of interviews) was determined. As outlined on page 5, rather than saturation, we sought breadth of perspectives and recruited sites and interviews accordingly within the parameters of the resources available. To facilitate comparison, we selected six sites (two-two-two), in line with Yin’s methodology – an approach widely recognised in case study research. We have noted the limitation of the study only including two sites per level of engagement with the GIRFT programme - see addition to discussion on page 15.
3. The purposive sampling seems to have been successful in reaching a range of hospital types and settings. What was the nature of the Trusts that were approached to participate but where engagement was not achieved? This should be discussed in the	Information summarising sites that did not take part has been included in the case selection section (see page 5) and a limitation added on page 15. In summary, we had a list of potential sites to recruit by level of engagement. We invited two

limitations, particularly if there is the possibility that negative perception or experience with GIRFT was a significant factor in the Trust's lack of interest in engagement in the research.	case sites per level of engagement and continued to approach Trusts until we recruited two sites per level of engagement. Two sites did not respond to our invitations to be a case sites – one was highly engaged, and one had low engagement according to the GIRFT programme team's classifications. We do not know the reasons for their lack of interest in taking part in our study. Additional information has been provided in the methods section and in the limitations.
4. Little detail is provided on the analysis methodology. Were transcripts double coded and how were the individual researchers' analyses combined? The observation of the GIRFT visits seems a valuable part of the data collection but it is not clear what contribution to the analysis and findings this element provided. For example – the findings from the interviews include the statement that the GIRFT visits were not opportunities for two-way dialogue. Could the field journal notes be used to comment on this issue from the perspective of an observer?	Additional detail has been added to the manuscript about the stages of analysis - see page 7 The findings present an integration of interview and observation data (see page 7 comments about stages of analysis) and the data from observations confirm those about a lack of two-way dialogue. This clarification has been included on page 10.
5. Comment is made that the interviews took place several months after the GIRFT visits – could more detail be provided on the timings of the interviews? The data collection window is wide (Oct 2016 to May 2019).	Data collection timescales were led by the timing of GIRFT deep-dive visits to the case study sites. We are not able to provide detailed information about this in the manuscript because this might identify the case sites to the GIRFT team. However, we have clarified the reason for this wide data collection window in the manuscript – see page 6.
6. The quotes on page 10, lines 55 and page 11 line 7 are distracting due to the use of the square brackets to correct '[b]ecause', and 'vy[e]'. Are these necessary? were they from written documents or typing mistakes in transcription from audio recording? Vie, rather than vye?	The square brackets need to remain for 'Vy[e]' because we had to change the word ending (originally vying) to fit with the sentence structure in the manuscript, We have removed the square brackets in '[b]ecause', because this was a transcription editing decision.
7. The related paper: Barratt H, et al. BMJ Open 2022;12:e058316. doi:10.1136/bmjopen-2021-058316 suggests that "later cohorts had made changes prior to their visit because	We did not explore this in the qualitative interviews because the quantitative analyses which led to positing of this as a possible explanation for the trend results were completed after the qualitative data collection ended. No

of information they gleaned from Trusts visited earlier in the process” was this explored in this qualitative work?	changes have been made to this manuscript in response to this comment.
8. Could the authors provide a view on researcher reflexivity in this study? (as per the SRQR supplementary document). This may need to take a particular focus on the impact of having observed the visits themselves ahead of the interview process. Did researcher knowledge of the GIRFT programme’s categorisation of some of the sites as having had ‘low engagement’ shape the data collection and analysis?	A reflexive paragraph has been added to the discussion – see page 15
9. The findings present useful detail on the nature of the single clinical lead in this programme, the differences in perception of his style and contribution, and the potential impact of this role. These findings do not seem to have been explicitly translated into the recommendations; can the researchers provide any further recommendations specific to this insight?	We have added reflections on the leadership approach taken in the orthopaedic arm of the GIRFT programme - see page 14.
10. The consent process for the interviews is clear – detail is required on the consent situation for the visit observations.	Information about the consenting process, and limits to consent, has been added to the paper - see page 6.

VERSION 2 – REVIEW

REVIEWER	McLaughlin, Joanna University of Bristol, Translational Health Sciences
REVIEW RETURNED	03-Feb-2023
GENERAL COMMENTS	Dear authors, Thank you for the revised version of this manuscript. I am satisfied that my comments from my initial review have all been comprehensively addressed and I have no further suggestions for further revisions.